# Spatial Characteristics of Wildlife-Vehicle Collisions of Water Deer in Korea Expressway

**Hyomin Park, Minkyung Kim and Sangdon Lee ***

Department of Environmental Science & Engineering, College of Engineering, Ewha Womans University, Seoul 03760, Korea; hyomin@ewhain.net (H.P.); enviecol@ewha.ac.kr (M.K.)
* Correspondence: lsd@ewha.ac.kr; Tel.: +82-232-773-545; Fax: +82-232-773-275

**Abstract:** In recent decades, rapid industrial growth has accelerated the construction of new roads, which has led to the destruction and isolation of wildlife habitats. Newly constructed roads affect wildlife in many ways. In particular, fatal wildlife-vehicle collisions (WVCs) have a direct impact on wildlife. A substantial number of WVCs occur every year on expressways, where vehicle speeds and vehicle traffic are significant. However, our understanding of the relative importance of the factors associated with areas in which large numbers of WVCs occur on the expressway remains poor. Therefore, herein, we analyze the spatial characteristics of WVCs. The effect of spatial distribution on the occurrence of WVCs was analyzed using the types of land cover in the areas where water deer appear (Cheongju, Boeun, and Sangju) and the areas in which WVCs occur along the Cheongju–Sangju Expressway (CSE). We identified the WVC hotspots by using CSE patrol data recorded between January 2008 and December 2019, and we analyzed the corresponding distribution patterns and land cover characteristics. Along the CSE, a total of 1082 WVCs occurred, out of which collisions involving water deer (*Hydropotes inermis argyropus*) accounted for 91%. Water deer appear frequently in Forested Areas and Agricultural Land, but the WVC distribution in the Hotspots followed a highly clustered pattern, with a higher proportion of WVCs occurring in Used Areas (areas including buildings such as residential facilities, commercial and industrial facilities, and transportation facilities). Used Areas have a smaller cut slope compared to Forested Areas, and Used Areas are open terrains. Therefore, the occurrence of WVCs will be high given that wildlife can easily access the expressway. Based on these results, we can infer that the landscapes near the expressway influence the occurrence of WVCs. To establish an effective policy for reducing WVCs on a road, the WVC characteristics and spatial distribution of the road should be considered together. Further research on the wildlife ecology and land-use status of WVC hotspots is required to mitigate WVCs on expressways and protect human and animal life. Therefore, if the characteristics of WVC hotspots are analyzed considering the characteristics of various ecosystems, an appropriate WVC reduction plan can be established.

**Keywords:** wildlife vehicle collisions; roadkill; expressway; water deer; kernel density estimation; nearest neighbor method



## 1. Introduction

Roads affect wildlife ecosystems in various ways, including changing the land use patterns of animals, altering their behavioral patterns [1,2], and affecting dynamic community structures and components of ecosystems [3,4]. These factors influence the sustainability of animal species [5,6]. New roads affect wildlife in many ways, for example, they fragment communities and form barriers to movement [7–10]. Among these, fatal animal-vehicle collisions have direct impact on wildlife [7,8]. This phenomenon, called wildlife-vehicle collisions (WVCs), poses a serious threat to wild animals by endangering numerous animal species and also to vehicle drivers by causing loss of human life or property [11]. The global data on wildlife killed in WVCs shows that the species (Mammals) that crossroads have a higher probability of being killed while crossing than other species (birds, reptiles,

amphibians) [12–14]. It is estimated that more than a million vertebrates are involved in fatal WVCs per day in the U.S. [15]. On expressways, where vehicle speed and vehicular traffic are greater than on other roads, the occurrence of roadkill and isolation of animals are especially frequent [16]. Expressways are wider than other roads, and their proximity to forests significantly increases the risk of WVCs on them compared to that on other roads. As described above, expressways pose threats to animals. However, industrial development is stimulating the construction of new expressways, which in turn, would expectedly increase the occurrence of WVCs.

The rising social and economic costs of WVCs have prompted researchers to conduct cost-benefit analyses on mitigation strategies [17,18]. The construction of wildlife crossings, designed to connect wild animal habitats divided by expressways, is one of the most well-known mitigation measures [19,20]. Wildlife crossings for ungulates and faunae are located below or above expressways, and they aim to provide safe passage to these types of animals. Although WVC mitigation measures such as wildlife crossings and fences are highly effective at reducing WVCs, their construction warrants considerable investment [11]. Additionally, these mitigation measures tend to rely on past experiences rather than empirical analysis. Therefore, it is necessary to rank WVC reduction measures in terms of their effectiveness to establish the priority of WVC reduction measures by performing a quantitative analysis of the causes of WVC occurrence.

Compared to past studies on WVC that have mainly addressed the status of WVCs, many recent studies have focused on analyzing the causes of WVCs and developing approaches to mitigate them. These studies include analyses of WVC occurrence depending on landscape variables [21–23], studies on the social costs of WVCs [11,17], and research verifying the effects of WVC mitigation strategies [24]. In the extensive literature on WVC occurrence, many studies on the correlation between WVC occurrence and landscape variables have been conducted worldwide [21,22], but such studies have been rarely conducted in Korea. Therefore, it is necessary to analyze the correlation between WVC occurrence and landscape by using Korea's WVC occurrence data to formulate a WVC reduction plan.

Animals are mainly found near their shelters and near locations where food is readily available. Expressways are built through wildlife habitats. Therefore, wildlife will cross any expressway built within their home range, which is the primary cause of WVCs. For this reason, it is extremely important to study the relationship between the landscape around WVC hotspots and the home ranges of wildlife.

In this study, we obtain geographical information regarding animal migration based on WVC occurrences on the expressway and designated WVC hotspots. A "hotspot" is an area in which the occurrence risk of WVCs is statistically higher than that in other regions [25]. It is vital to identify these hotspots for reducing collision-related threats to wildlife [26,27]. Research on WVC hotspots involving wild animals has been widely used to prioritize locations for WVC reduction [22]. It is essential to identify spatial and temporal hotspots and analyze their effects on WVC occurrence to establish mitigation measures and thus avoid high WVC.

Insufficient data regarding wildlife distribution in a given area can limit one's ability to predict the occurrence of WVCs in the area. Therefore, uniformly collected WVC data are extremely important for WVC prediction. In addition, analysis conducted using a WVC occurrence map may not precisely indicate the accident concentration, especially at spots where more than two WVCs have occurred. Therefore, it is necessary to analyze the density of WVC occurrence at the same spot. This will help to predict the occurrence of WVCs more accurately and to establish an effective WVC reduction plan.

To this end, in the present study, we aim to understand the landscape characteristics of the locations frequented by water deer and the locations of WVC occurrence. Then, we identify WVC hotspots and analyze their spatial characteristics. Finally, we analyze the WVC cluster characteristics of the identified WVC hotspots. In doing so, we ana-

lyze the spatial characteristics of WVC occurrence to gather the basic data necessary for implementing WVC reduction plans on expressways.

## 2. Materials and Method

### 2.1. Study Area

The study area was the Cheongju–Sangju Expressway (CSE), which connects the inland provinces of Cheongju-si, Chungbuk-do, Boeun-gun, Chungbuk-do, Sangju-si, and Gyeongbuk-do (Figure 1). This expressway is a portion of the Dangjin–Yeongdeok Expressway (total length = 278.6 km), and it was completed in November 2007. The full length of the expressway is 79.4 km, and it is a two-lane dual carriageway. The maximum and minimum speed limits on this expressway are 110 km/h and 50 km/h, respectively.

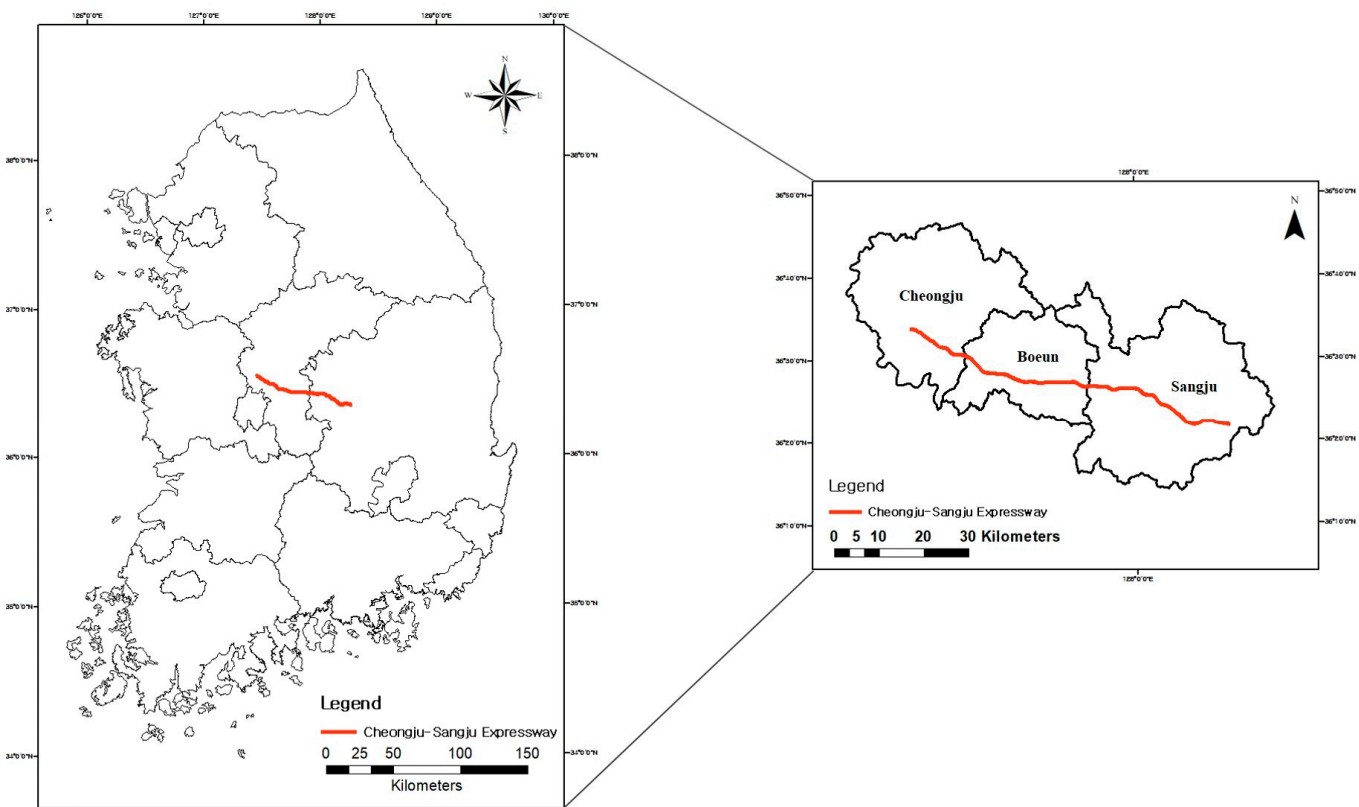

**Figure 1.** Map of the study area in South Korea showing the sections of the Expressway that were surveyed. The red line represents the Cheongju–Sangju expressway (CSE).

The CSE is located near the Songnisan National Park and so passes through abundant forests and various types of land cover. Moreover, the CSE is a relatively recent addition to the Korean Expressway system built over the home range of wildlife. Therefore, the possibility of WVC occurrence is high compared to other expressways. We selected this region as the study area because it provides ample opportunities to compare animal habitats and WVC occurrence points in a region where WVCs occur frequently.

### 2.2. Data Collection

The Korea Expressway Corporation provided the WVC occurrence data for the section of the CSE considered in this study. The analysis was conducted based on WVC occurrence on the CSE from the time it was inaugurated in January 2008 to December 2019 (12 years in total). The total number of WVCs on the CSE during this period was 1082 (Table 1). The majority of the WVCs involve water deer (*Hydropotes inermis argyropus*) (91.04%). Therefore, we conducted the analysis by using the WVC data of water deer.

The Korea Expressway Corporation employees in charge of WVCs record WVC data on a daily basis. These WVC data, including the number and type of carcasses discovered during daily expressway patrols in three shifts, are collected in the form of field notes by the Safety Patrols of the Korea Expressway Corporation. In the field notes, the patrol personnel note down WVC dates, times, the authority having jurisdiction, route name, direction, species of animal killed in a WVC, and longitude and latitude of WVC occurrence (http://data.ex.co.kr/ (accessed on 29 May 2020)). Thereafter, they remove the animal carcasses from the road to ensure that the same carcass is not recorded twice.

**Table 1.** Number of wildlife-vehicle collisions (WVCs) on Cheongju–Sangju Expressway (CSE) during 2008–2019.

| Species | WVCs | Percentage (%) |
|---|---|---|
| Water deer (*Hydropotes inermis*) | 985 | 91.04 |
| Raccoon dog (*Nyctereutes procyonoides*) | 43 | 3.97 |
| Wild boar (*Sus scrofa*) | 14 | 1.29 |
| Leopard cat (*Prionailurus bengalensis*) | 14 | 1.29 |
| Asian badger (*Meles leucurus*) | 14 | 1.29 |
| Korean hare (*Lepus coreanus*) | 6 | 0.55 |
| others | 6 | 0.55 |
| Total | 1082 | 100 |

*2.3. Study Species*

Water deer, the most common victim of WVCs on the CSE, was selected as the primary focus of this research. It may not be possible to find the carcasses of smaller animals, such as amphibians and reptiles, on the expressway compared to those of large mammals. WVCs involving smaller animals are often underreported relative to the ones involving large animals [19,20]. Such inaccuracies can distort the WVC Hotspot results [28]. Therefore, we have selected water deer, which account for the majority of WVCs on the CSE and there is a low risk of underreporting of WVCs involving this species.

Water deer are indigenous only to Korea and China. The distribution of Korean water deer is widespread [29], and its inhabitation density has been increasing (from 2.4 per 100 ha in 1989 to 7.9 per 100 ha in 2019). The habitats of water deer are widely distributed, except in island areas [30]. In Korea, water deer are designated harmful wildlife because of their high regional habitat density and hunting of water deer has been permitted since 2005.

*2.4. Data Analysis*

2.4.1. Spatial Analysis Using Land Cover

The difference in spatial characteristics between the areas where water deer appear and the areas where WVCs occur were compared. Data from the National Ecology Survey were used for the area in which water deer appear. We analyzed the land cover over a radius of 500 m from the point at which water deer appeared and the point at which WVCs occurred; then, we compared the results.

While there is no research available in Korea on the spatial range over which water deer are affected, research on the behavior of mule deer (*Odocoileus hemionus*) suggests that ungulates are affected over the spatial range of 300 m from the road [31,32]. As a result of studying the movements of water deer by attaching a GPS to four water deer, it was found that the home range of water deer was 0.24 km$^2$ during the day and 0.26 km$^2$ at night [33]. Considering these data, the radius of the home range of water deer was set to 500 m in our analysis.

The National Ecosystem Survey, conducted by the Ministry of Environment, is the largest national natural environment survey project is implemented throughout the country and investigates the entire natural ecosystem, including vegetation, flora, fauna, and topography. The national ecosystem survey data can be obtained from www.nie-ecobank.kr. Among the published 2nd (1997–2005), 3rd (2006–2013), and 4th (2014–2018) national

ecosystem survey data, the data on water deer that appeared in Cheongju, Boeun, and Sangju were extracted.

The land cover data used in this study were recorded in the late 2010s. There are seven categories for land cover: Used Area (areas including buildings such as residential facilities, commercial and industrial facilities, and transportation facilities), Agricultural Land (agricultural areas for farming, areas for growing fruit trees and street trees, and facilities used for livestock and dairy farming), Forested Area (area in which trees grow in groups), Grassland (land covered with herbaceous plants, including both natural grassland and artificial grassland), Wetland (wet soil where moisture is always maintained by the natural environment), Barren Land (bare ground without vegetation cover), and Water (low areas with stagnant water, such as lakes, reservoirs, and swamps). The land cover map was obtained from https://egis.me.go.kr/ (accessed on 5 March 2020), which is operated by the Ministry of Environment.

The above analyses were performed using Arc GIS version 10.7.1. The spatial correlation between the appearance of water deer and the points of WVC occurrence was determined by conducting a Spearman's correlation analysis. Statistical analysis was performed using SPSS 25.0 version.

### 2.4.2. WVC Hotspot Modelling

Kernel density estimation (KDE), a technique to determine point density, was used to evaluate WVC Hotspots along the CSE. KDE can be used to estimate the spatial density of the entire area based on the distribution of point features [22]. This technique is widely used to visualize the spatial distribution patterns of point data because it helps with obtaining a conceptual understanding and interpreting spatial densities. Specifically, it can be used to understand the degree of clustering of point features based on hotspots [34]. By using KDE, we classified WVC occurrence density on the CSE into five sections (I, II, III, IV, V) with the same interval values and identified WVC hotspots (Ranges of the sections—I: 80–100%, II: 60–80%, III: 40–60%, IV: 20–40%, and V: 0–20%). Section I covers the areas with extremely high WVC occurrence densities (top 20%). By contrast, Section V covers the areas with extremely low or zero WVC occurrence density (bottom 20%).

After identifying the sections with WVC hotspots, the study area and a land cover map recorded in the late 2010s were overlaid using Arc GIS ver.10.7.1. After analyzing the land cover characteristics of each section, the land cover pattern of the WVC occurrence area was analyzed. Two independent sample t-tests were conducted to analyze the statistical significance of the differences in land cover types between the WVC hotspot and non-hotspot sections. Statistical analysis was performed using SPSS version 25.0.

### 2.4.3. WVC Occurrence Cluster Analysis

The nearest neighbor method (NNM) relies on the distance between the nearest neighbors that are the closest to a certain point [35,36]. If the nearest neighbor ratio (NNR) equals 1, the distribution is completely random. By contrast, an NNR value greater than 1 indicates a dispersed distribution, and an NNR value less than 1 indicates a clustered distribution [37]. Based on section analysis conducted using KDE, the degree of WVC clustering, the WVC Hotspots, and each section (I–V) was analyzed using the NNM. Then, the characteristics of the WVC occurrence clusters in each section were analyzed.

## 3. Results and Discussion

### 3.1. Spatial Analysis Using Land Cover

To determine the land cover in the area where water deer exhibit active behavioral patterns, the points of appearance of water deer, as recorded in the national ecosystem survey, were used (Table 2). The results of our analysis of the land cover around the locations at which water deer were found in Cheongju, Boeun, and Sangju based on the data from the National ecology survey indicated that Forested Areas showed the

highest ratio (79.17%), followed by Agricultural Land (12.44%), Grassland (2.84%), and Water (2.02%).

The land cover of the areas with WVC occurrence was analyzed and compared with the land cover in the areas where water deer appeared. The type of land cover in areas close to the points of occurrence of WVCs indicated that Forested Areas comprised the majority of the WVC areas (69.72%), followed by Agricultural Land (17.92%), Used Area (6.19%), and Grassland (4.09%).

**Table 2.** Result of spatial analysis conducted using land coverage maps obtained from National Ecological Survey and data on the occurrence of wildlife-vehicle-collisions (WVCs) on CSE.

| Land Cover Pattern | National Ecosystem Survey (%) | WVCs (%) |
|---|---|---|
| Used Area | 1.75 | 6.19 |
| Agricultural Land | 12.44 | 17.92 |
| Forested Area | 79.17 | 69.72 |
| Grassland | 2.84 | 4.09 |
| Wetland | 1.02 | 0.39 |
| Barren Land | 0.75 | 0.99 |
| Water | 2.02 | 0.71 |
| Total | 100 | 100 |

A comparison of the land cover in the areas where water deer appeared and the land cover in the areas where WVCs occurred revealed that the types of land cover in the areas where water deer frequently appeared were Forested Area and Agricultural Land. Moreover, the types of land cover in the areas where WVCs occurred were Forested Area, Agricultural Lands, and Used area, which were similar to the land cover results of the areas in which water deer appeared in the natural ecosystem survey.

The home range of water deer is Forested Areas, Agricultural Land, and Wetland [33]. In addition, the results of previous studies indicated that Forested areas and Agricultural Land tended to increase the occurrence of WVCs [38]. Given that the land cover in the areas of WVC occurrence is similar to the habitat range of water deer, the appearance probability of water deer is high. This increases the occurrence probability of WVCs.

The density of WVCs per unit area (km$^2$) for the types of land cover around the CSE was analyzed (Table 3). The results indicate that the density of WVCs in Used Areas was the highest (27.41 cases per km$^2$), followed by Wetland (12.57 cases per km$^2$), Forested Areas (10.90 cases per km$^2$), and Agricultural Lands (10.58 cases per km$^2$). These results suggest that there is a difference between the appearance of water deer and the type of land cover where WVCs occur frequently.

**Table 3.** WVCs density obtained by calculating WVC occurrence per km$^2$ of land cover by classifying WVC occurrence along CSE routes by land cover. Based on the land cover in the study area, the relative ratios of land cover in areas with water deer appearance (National Ecosystem Survey) and land cover in areas with WVC occurrence were analyzed. If the result is greater than 1, the cover value is higher than the land cover ratio of the study area; if the result is less than 1, the cover value is lower than the land cover ratio of the study area.

| Land Cover Pattern | WVCs Density(/km$^2$) | Relative Land Cover Ratio of the Study Area | |
|---|---|---|---|
| | | National Ecosystem Survey | CSE |
| Used Area | 27.4 | 0.41 | 1.37 |
| Agricultural Land | 9.94 | 0.75 | 1.15 |
| Forested Area | 10.36 | 1.12 | 0.97 |
| Grassland | 7.25 | 0.69 | 1.01 |
| Wetland | 12.04 | 1.11 | 0.51 |
| Barren Land | 4.28 | 0.64 | 0.92 |
| Water | 4.57 | 0.89 | 0.33 |

Therefore, the differences in spatial characteristics between the areas where water deer appeared (National Ecology Survey) and the areas where WVCs occurred were compared. Based on the land cover in the study area (Cheongju, Boeun, and Sangju), the relative ratios of land cover in areas with water deer appearance (National Ecosystem Survey) and land cover in areas with WVC occurrence were analyzed (Table 3).

As of 2010, the types of land cover in all Korean territories were as follows: Forested Areas (67.8%), Agricultural Lands (21.1%), Used Areas (4.1%), and Grassland (2.9%) [39]. The land cover across Korea and the land cover in the area where water deer appeared were compared. The results indicated that the land cover ratios of Forested Area, Grassland, Wetland, and Bare Land in the areas where water deer appeared were higher than the corresponding ratios for all of Korea. The land cover ratios of Forested Area, Grassland, and Wetland in the areas where water deer appear are higher than those of Korea because CSE is located near Songnisan National Park. For this reason, there is a high probability that a larger number of water deer will inhabit the study area compared to other areas in Korea. In previous studies, the higher the habitat quality, the more vulnerable the area was to WVCs occurring near expressways [22]. Therefore, the study area has a favorable environment for water deer to inhabit, and it has a high occurrence probability of WVCs.

The area in which water deer appeared had a higher ratio of Forested Area and Wetland compared to the study area, but the ratio of Used Area was lower. According to the present study, water deer frequent Forested Areas and Wetland, and this finding is consistent with the results of previous studies [33] on the habitats of water deer. However, in the areas with WVC occurrence, the land cover ratios of Used Areas, Agricultural Land, and Grassland were higher than those in the study area. Used Area, Agricultural Land, and Grassland are areas characterized by low elevation and high openness. In previous studies, water deer had a high habitat density at elevations lower than 300 m [40]. Therefore, the density of water deer is expected to be high in Used Area, Agricultural Land, and Grassland. Used Area and Grassland are wide and open terrains, so water deer can access them easily.

Based on the results, correlations with land cover in the areas where water deer appeared around the CSE and the areas in which WVCs occurred were statistically analyzed by conducting Spearman's correlation test. As a result, the correlation of land cover between the areas where water deer appeared and the areas where WVCs occurred was −0.535, which is a negative correlation.

### 3.2. WVC Hotspot Modelling

Based on an analysis of the location data of WVC occurrence with KDE, WVC occurrence density was divided into five sections (Table 4, Figure 2). In Section I, which had the highest WVC occurrence density, 35.49 cases of WVC occurred per km, while 23.98 cases occurred per km in Section II. By contrast, the number of WVC occurrences per km in section V, which had the lowest WVC density, was 1.75.

The sections that exhibited higher WVC densities (cases per km) than that of the CSE overall (12.36 cases per km) were defined as WVC hotspots, and these hotspots were located in Sections I and II. The WVC density in the hotspots was 26.96 cases per km. The occurrence of WVCs in Section I was 20.28 times higher than that in Section V. Section I accounted for 5.30% of the CSE and 15.23% of the WVCs occurring along the entirety of the CSE. Section II accounted for 15.23% of the CSE and 29.54% of WVCs along the CSE. Based on these values, it can be concluded that even though the WVC Hotspots merely accounted for 20.53% of the CSE study route, the WVC occurrence ratio in the corresponding sections was 44.77%. These statistics indicate that most of the WVCs on the CSE occurred in Sections I and II. The WVC occurrence densities in Sections I and II were 26.31 per $km^2$ and 21.24 per $km^2$, respectively. A high percentage of the WVCs occurred in Sections I and II, and the WVC density in Section I was 12.96 times higher than that in Section V.

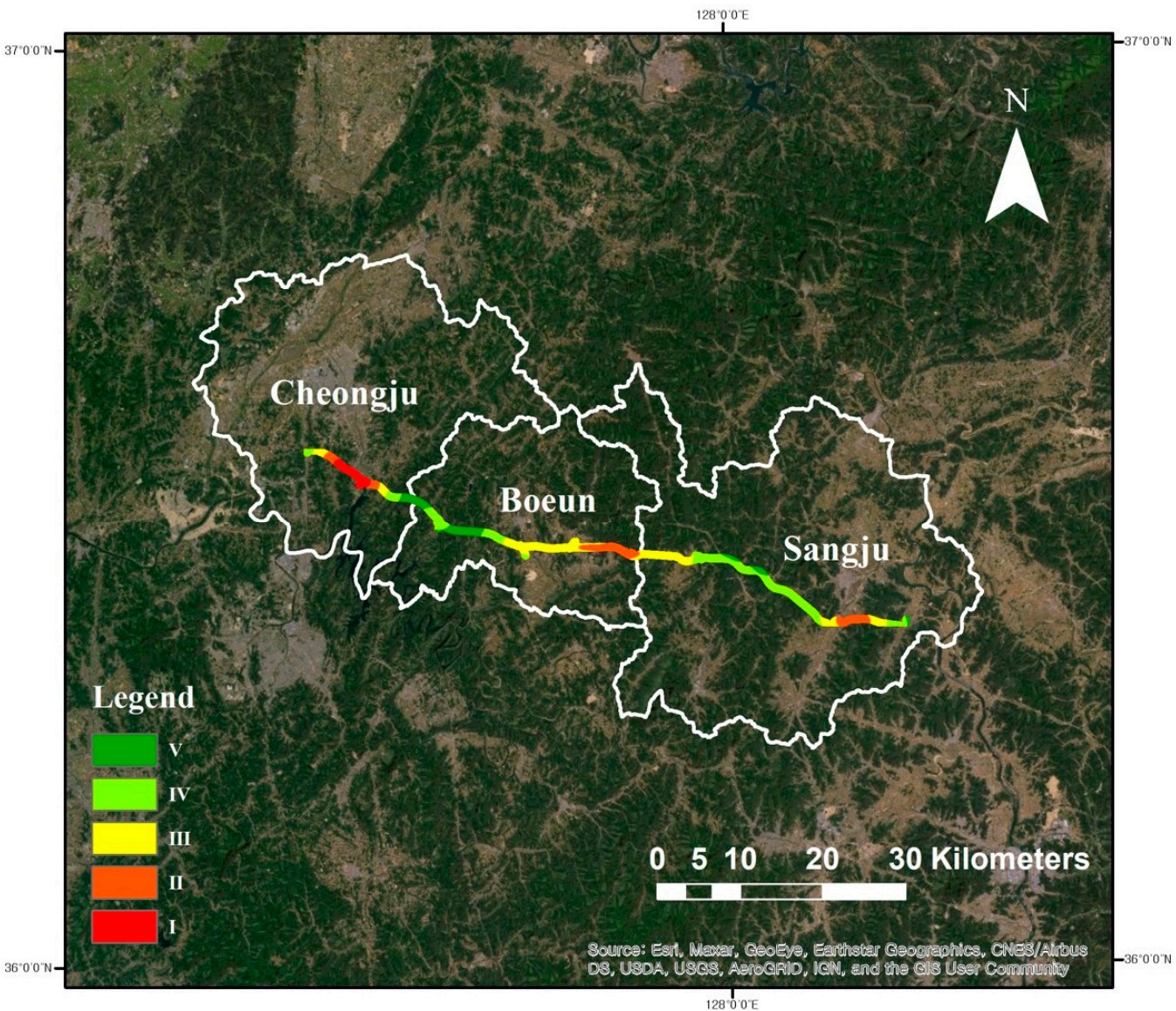

**Figure 2.** The result of dividing the probability of water deer wildlife-vehicle collision (WVC) occurrence density on the Cheongju–Sangju Expressway (CSE) into five sections by using the kernel density estimation (KDE) technique. Sections with high probability of WVC occurrence density are marked in red (I), and sections with low probability of WVC occurrence density are marked in green (V). The WVC occurrence density along the CSE was classified into five sections (I, II, III, IV, and V) with the same interval values (Ranges of the sections—I: 80–100%, II: 60–80%, III: 40–60%, IV: 20–40%, and V: 0–20%).

Based on the probability of WVC density on the CSE, each section was overlaid with a land cover map to analyze the land cover ratio of each KDE-derived section. The land cover ratios in the areas with WVC Hotspots were as follows: Forested Area = 65.33%, Agricultural Land = 19.58%, and Used Area = 7.06%. Compared to the WVC occurrence points (Table 2), WVC hotspots had low ratios of Forested Area. Compared to the land cover in areas with water deer appearance (National Ecosystem Survey), the ratios of Used Area, Agricultural Land, Grassland, and Barren Land in the WVC Hotspot areas were high.

The density of WVC occurrence by land cover was analyzed (Table 5). As a result, it was found that along the entire CSE and in the areas with WVC Hotspots, the highest WVC occurrence density was in the Used Area. The WVC density in the WVC hotspots generated in the Used Area was three times higher than that in the Forested Area. The density of WVC by land cover was highest in the Used Area, except in Section I.

**Table 4.** Analysis of wildlife-vehicle collision (WVCs) density along the Cheongju–Sangju Expressway (CSE) by using the kernel density estimation (KDE) technique and the current status of WVCs in the sections divided based on WVC density; the result of analysis of the characteristics of land cover types in each section by using land cover maps of the five sections classified using the KDE technique.

|  |  | I | II | III | IV | V | All Routes | Hotspot (I + II) |
|---|---|---|---|---|---|---|---|---|
| Land cover value (%) | Used Area | 7.23 | 6.98 | 5.91 | 5.76 | 3.55 | 5.82 | 7.06 |
|  | Agricultural Land | 19.10 | 19.78 | 26.39 | 15.87 | 7.24 | 18.74 | 19.58 |
|  | Forested Area | 64.51 | 65.66 | 60.30 | 72.80 | 87.04 | 69.20 | 65.33 |
|  | Grassland | 8.10 | 3.86 | 4.84 | 3.34 | 1.60 | 3.97 | 5.10 |
|  | Wetland | 0.08 | 0.68 | 0.79 | 0.32 | 0.07 | 0.47 | 0.50 |
|  | Barren Land | 0.87 | 2.27 | 1.23 | 0.60 | 0.39 | 1.05 | 1.86 |
|  | Water | 0.11 | 0.76 | 0.53 | 1.32 | 0.12 | 0.76 | 0.57 |
| Km |  | 4.23 | 12.13 | 24.58 | 26.16 | 12.6 | 79.70 | 16.36 |
| WVCs |  | 150 | 291 | 302 | 220 | 22 | 985 | 441 |
| WVCs/km |  | 35.49 | 23.98 | 12.28 | 8.41 | 1.75 | 12.36 | 26.96 |
| WVCs density (/km$^2$) |  | 26.31 | 21.24 | 11.92 | 7.40 | 2.03 | 11.55 | 22.73 |

**Table 5.** Result of analysis of the density of Wildlife Vehicle Collision (WVCs) occurrence by land cover in the five sections classified using the Kernel Density Estimation (KDE) technique, all Cheongju–Sangju Expressway (CSE) routes, and WVC hotspot sections.

|  |  | I | II | III | IV | V | All Routes | Hotspot (I + II) |
|---|---|---|---|---|---|---|---|---|
| WVC density (/km$^2$) | Used Area | 21.83 | 81.53 | 19.36 | 10.52 | 5.20 | 27.41 | 63.55 |
|  | Agricultural Land | 24.79 | 20.66 | 7.78 | 7.21 | - | 10.58 | 21.85 |
|  | Forested Area | 30.45 | 17.01 | 13.10 | 7.31 | 2.12 | 10.90 | 20.91 |
|  | Grassland | 4.33 | - | 13.86 | 6.04 | - | 7.39 | 2.02 |
|  | Wetland | - | 10.79 | 20.02 | - | - | 12.57 | 10.29 |
|  | Barren Land | - | 9.63 | - | 5.61 | - | 4.48 | 8.31 |
|  | Water | - | - | - | 7.66 | - | 4.61 | - |
| Total |  | 26.31 | 21.24 | 11.92 | 7.40 | 2.03 | 11.55 | 22.73 |

Since WVCs are easily affected by land cover [41], we found it necessary to compare the land cover maps of WVC Hotspots and other sections on the CSE. The land around Sections I and II had lower percentages of Forested Areas than the land in Sections III, IV, and V, and higher percentages of Used Area and Grassland. This result is consistent with that of a previous study on the suitability of water deer habitats with MaxEnt, which suggested that water deer prefer Used Areas among the different land cover types [42]. The WVC Hotspots were characterized by higher percentages of open terrains such as Used Area, Grassland, and Barren Land. The open terrains, where roads are located, have fewer cut slopes than Forested Areas, which allows animals to easily access the expressway and increases the occurrence rate of WVCs [43]. Water deer, which especially prefer to live on the edges of forests, tend to appear more frequently in Grasslands than in Forested Areas. The WVC occurrence density results provide evidence that WVC cases occur more frequently in open terrains [43,44]. Moreover, according to the field survey results, the closer a facility is to a Used Area, the higher is the occurrence probability of WVCs [45] because in these areas, food is easy to find, and the occurrence probability of WVC is high due to the flat terrain.

Meanwhile, Sections IV and V, where WVC densities involving water deer were relatively low, had higher ratios of Forested Areas. A forested land cover and proximity to roads are spatial factors that can predict the WVCs of different ungulates (roe deer in Austria and France and white-tailed deer *Odocoileus virginianus* in Illinois and Pennsylvania) [46–50]. For instance, as the distance between a forest and a road increased by 100 m, the collision risk with moose decreased by 15% [49]. In a previous study, it was demonstrated that forested habitats, which provide food sources to wildlife, and proximity to roads significantly influence the occurrence risk of WVCs [49]. Thus, Sections IV and V, which had higher ratios of forests with trees and herbaceous plants, provided

abundant food sources and resting places for water deer, such as lairs. Hence, in the areas with forested habitats, fewer WVCs involving water deer occurred compared to those in Sections I and II.

Two independent sample t-tests were conducted to analyze the statistical differences between the land covers in the WVC hotspots (Sections I and II) and non-WVC hotspots (Sections III, IV and V). The results showed a statistically significant difference (t = 5.655, significance level of 0.000, $p < 0.05$) in the land cover of the WVC Hotspot.

### 3.3. WVC Occurrence Cluster Analysis

The WVC distribution pattern in each of the five sections classified using the KDE function based on the WVC occurrence density probability was analyzed using the NNM (Table 4). The distribution pattern in Section II was the most clustered with an NNR of 0.046 while that in Section V was the most dispersed with an NNR of 1.950 (Table 6).

**Table 6.** Analysis result of the distribution pattern of wildlife-vehicle collisions (WVCs) occurring in each section by using the nearest neighbor method after dividing the WVCs that occurred on the Cheongju–Sangju Expressway (CSE) into five sections based on their probabilities of occurrence density.

|  | **Nearest Neighbor Ratio** | **WVC** |
|---|---|---|
| I | 0.086 | 150 |
| II | 0.046 | 291 |
| III | 0.050 | 302 |
| IV | 0.098 | 220 |
| V | 1.950 | 22 |
| All routes | 0.097 | 985 |
| Hotspot I + II | 0.043 | 441 |

The WVC distribution patterns in the WVC Hotspots, entire CSE, and five sections analyzed using the KDE technique were examined using the NNM. While the CSE as a whole had an NNR of 0.097, the WVC hotspots had an NNR value of 0.043. These NNR values indicate that the WVC occurrence pattern in the hotspots followed a highly clustered distribution compared to that along the entire expressway. The distribution in Section I was relatively dispersed compared to those in Sections II and III. The WVC cases occurring in Section I were distributed over the entirety of Section I. The WVC Hotspots had high distributions and densities of wildlife because they were the transit regions for wildlife [51,52]. Section 1, as classified using KDE, accounted for 5.30% of the CSE, but the density of WVC occurrence in this section was high. Since the occurrence probability of WVCs was high throughout Section I, a distributed distribution was observed in Section I.

The NNR values of Sections II and III were 0.046 and 0.050, respectively, indicating clustered distributions. In Section II, the density of WVC occurrence in the Used Area was the highest at 81.53 cases per km$^2$ (Table 5), but the land coverage ratio of Used Area was only 6.98% (Table 4). The density of WVC occurrence in the Used Area in section II was 3.83 times higher than that in the entirety of section II. Therefore, the occurrence of WVCs was high in a small patch of Used Area, which implies that WVC occurrence in the section followed a clustered distribution. As for the land cover in Section III, the proportions of Forested Area and Agricultural Land were extremely high, accounting for 86.69% of the total land in Section III (Table 4). However, the types of land cover with high WVC densities were Wetland and Used Area. Wet Land and Used Area had small distributions in Section III. Therefore, WVC occurrence in Section III tended to follow a clustered distribution compared to the other sections.

### 4. Conclusions

Water deer are exposed to many risks due to expressway construction, and for this reason, they are involved in WVCs. Most of the WVCs that occurred on the CSE involved

water deer. Therefore, we identified WVC hotspots on the CSE and investigated the effect of spatial distribution on WVC occurrence based on land cover in the areas where water deer appeared and land cover in the areas where WVCs occurred.

Most of the lands in the areas where water deer appears were Forested Areas and Agricultural Lands. The land cover ratios of Forested Area and Wetland in the areas where water deer appeared were higher than the corresponding values in the study area (Cheongju, Boeun, and Sangju), whereas the land cover ratio of Used Area was lower than that in the study area. In the areas with WVC occurrence, the land cover ratios of Used Area and Agricultural Land were higher than those in the study area.

An analysis of the occurrence density of WVCs indicated that many WVCs occur in Used Areas. Moreover, the density of WVC hotspots was the highest in Used Areas. Through cluster analysis, it was confirmed that rate of occurrence of WVCs was high in Used Areas. Used Areas have a smaller cut slope than Forested Areas, meaning that wildlife can easily access the expressway, and these results are consistent with those of a previous study, where it was reported that WVCs occur frequently on open terrains.

Furthermore, the occurrence of WVCs was found to be influenced by the spatial distribution around the CSE, and the WVC occurrence pattern tended to be spatially clustered. WVC occurrence followed a clustered distribution rather than a random pattern [49]. Moreover, WVC occurrence points were affected by the local circumstances and local landscapes [50]. Similar results were reported in other studies conducted by road ecologists, indicating that WVC occurrence patterns tend to be spatially clustered instead of random [26,51–53]. Therefore, to establish an effective policy for reducing WVCs on a road, the WVC characteristics and the spatial distribution of the road should be considered together.

WVCs have a significant influence on the safety of humans and animals. Therefore, it is essential to analyze the exact location, period, and causes of WVCs in high-risk areas and establish effective mitigation measures [54]. To reduce the probability of WVC occurrence on expressways, the locations of WVC mitigation facilities should be selected based on analyses of each animal's behavioral and ecological characteristics. New mitigation measures should address the factors affecting the spatial distribution of WVCs.

Herein, we quantitatively analyzed the occurrence of WVCs and studied the spatial characteristics that affect WVC occurrence. Therefore, the results of this study provide a quantitative priority of various landscapes for establishing a WVC reduction plan in the future. A limitation of this study is that we did not consider the characteristics of various ecosystems because we conducted the study on water deer without focusing on the underreporting of WVC data, and land cover was used among various spatial characteristics. Therefore, if the characteristics of WVC hotspots are analyzed considering the characteristics of various ecosystems in the future, an appropriate WVC reduction plan can be established.

**Author Contributions:** Data curation, H.P.; Formal analysis, H.P. and M.K.; Project administration, M.K.; Supervision, S.L. All authors have read and agreed to the published version of the manuscript.

**Funding:** The financial support was from KRF-2021R1A2C1011213 and MOE-2020002990006. The authors also wish to thank the Korea Expressway Corporation for providing data.

**Institutional Review Board Statement:** Not applicable.

**Informed Consent Statement:** Not applicable.

**Data Availability Statement:** Not applicable.

**Conflicts of Interest:** The authors declare no conflict of interest.

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
