# Peer review of "Spatial Characteristics of Wildlife-Vehicle Collisions of Water Deer in Korea Expressway"

_sustainability, doi:10.3390/su132413523_

Round 1
Reviewer 1 Report
The MS requires more review effort than I can possibly give and I will focus my review on part of the MS (methods and results). In addition, I was not able to find line numbers which would have facilitate the communication. I strongly encourage authors to number lines of their MS before submission and editors to check that, before sending around to referees.
Overall, the authors need to be more rigorous and accurate in specifying their underlying hypothesis, the choice of analyses and how the data were generated.
A reported killed animal in the dataset is the result of a cascade of conditional probabilities: it has to be reported, to be found, to be searched, it has to be killed by a car, to cross the road, to be next to a road, and to be in its distribution range. For example, the seasonal pattern reported in figure 3 may be interpreted as variation in any of this probability (e.g., detection of dead animal may be easier in spring, traffic may be more important in spring, observed variations may only reflect variation in local abundance of dear next to the road, etc.).
The authors implicitly assume that most of these probabilities are independent of seasons and that variation in numbers of deaths is due to seasonal variation in individual dear mobility. It is fairly plausible but the MS would gain in strength if this was discussed more systematically stating the assumptions and their level of plausibility (e.g., “dead dear are big animals with a high and constant detectability”; “searching effort is organized to ensure homogeneity through time and all along the highway”, etc.).
The analysis of spatial variation is much more problematic since many of these probabilities may vary in space and not only the biological variable of interest (propensity of an individual dear to expose itself to a car accident). For this part, the MS has to be improved throughout methods and results to reach the request level of rigor for evaluation (and eventual publication).
One aspect to clarify is the choice of spatial resolution of the relationship between the location of death and local habitat. This is partly linked to the way dear choose to cross the road. This is treated under the paragraph entitled “2) WVC Hotspot Modelling and Land Use analysis of WVC Hotspot area”, page 4.
Given that an individual dear is present next to a road (500mx500m), will it cross anywhere along the road along these 500m or will it make up to a 500m detour to cross the road at a specific point? It is also unclear how this 500m distance was chosen since 1 km, 300m, 1-2 km are mentioned in the previous sentences. The authors have to state clearly how landscape is assumed to affect roadkill (favor large number of dears next to the road vs forces dear to cross at some specific point).
The sentence: “Similarly, the land cover near the expressway was analyzed up to an area of 500 m from the ends of the expressway to account for the water deer’s habitual range.” is also quite mysterious. What is the “habitual range”? It seems that the authors would like to make inference from the comparison of landcover along the expressway next to vs far from roadkill, but since the underlying hypothesis are not specified (e.g., no roadkill = no dear or no roadkill = no road crossing?), again it is difficult to evaluate the rational of these analyses.
Equally problematic is the use of the national ecosystem survey data. We have no clue how these data are collected (opportunistic report? Systematic survey accounting for habitat dependent dear detectability?). Hence, the kind of inference that can be made for these data cannot be evaluated, as well as what the authors are expecting with the comparison between these habitat distribution data far from highway and close to highway.
All these spatial data are then analyzed with various methods (section 3.2 and 3.3) and I started to be highly confused. I could not follow what was the added value (specific hypothesis tested) of each of them. It seems that the authors tried any methods they knew rather than choosing the appropriate method for testing a clearly stated hypothesis. The regression analysis at the end of page 9 with a F = 239.69, R2 = 0.98, is a complete mystery (and such unusually high value are highly suspicious).
Author Response
First of all, thank you for taking your valuable time to review this article.
Based on the advice of the reviewers, the article was carefully revised. Thanks for the advice.
Overall, the authors need to be more rigorous and accurate in specifying their underlying hypothesis, the choice of analyses and how the data were generated.
- Thank you for your advice. We have revised or added explanations for hypotheses, analysis methods, and data generation methods.
A reported killed animal in the dataset is the result of a cascade of conditional probabilities: it has to be reported, to be found, to be searched, it has to be killed by a car, to cross the road, to be next to a road, and to be in its distribution range. For example, the seasonal pattern reported in figure 3 may be interpreted as variation in any of this probability (e.g., detection of dead animal may be easier in spring, traffic may be more important in spring, observed variations may only reflect variation in local abundance of dear next to the road, etc.).
The authors implicitly assume that most of these probabilities are independent of seasons and that variation in numbers of deaths is due to seasonal variation in individual dear mobility. It is fairly plausible but the MS would gain in strength if this was discussed more systematically stating the assumptions and their level of plausibility (e.g., “dead dear are big animals with a high and constant detectability”; “searching effort is organized to ensure homogeneity through time and all along the highway”, etc.).
- Expressways are built over the existing habitats of wildlife. Therefore, we think that the occurrence of WVCs is closely related to the ecological characteristics of wildlife. The hypotheses of this study have been added to the introduction.
- WVCs are regularly logged on a daily basis by staff dedicated to WVC. Therefore, it is a very regular material. An explanation for this has also been added in 2.2. Data Collection
- We revised the details of data collection in 2.2. Data Collection (Line 108-, Line 114-):
“The WVC data was collected by the Safety Patrols of the Korea Expressway Corporation, who has recorded the number and information of carcasses discovered during the daily patrol of the highway in three shifts a day in a field note. "
"After recording, the animal carcasses will be removed from the road and will not be included again in the WVC data.
The WVC data of Korea Expressway Corporation is recorded daily by employees in charge of WVC. So the data is recorded regularly. Therefore, this study used WVC data of Korea Expressway Corporation."
- Unlike water deer, small animals may be under-reported, which may affect Hotspot results. Additional information about this has been added in 2.3. Study Species(Line 120) :
“Water deer (Hydropotes inermis argyropus), the most common victim of WVCs on the expressway, was selected as the primary focus of this research. Smaller animals, such as amphibians and reptiles, may not be able to find carcasses on the expressway compared to large mammals. ”
The analysis of spatial variation is much more problematic since many of these probabilities may vary in space and not only the biological variable of interest (propensity of an individual dear to expose itself to a car accident). For this part, the MS has to be improved throughout methods and results to reach the request level of rigor for evaluation (and eventual publication).
One aspect to clarify is the choice of spatial resolution of the relationship between the location of death and local habitat. This is partly linked to the way dear choose to cross the road. This is treated under the paragraph entitled “2) WVC Hotspot Modelling and Land Use analysis of WVC Hotspot area”, page 4.
Given that an individual dear is present next to a road (500mx500m), will it cross anywhere along the road along these 500m or will it make up to a 500m detour to cross the road at a specific point? It is also unclear how this 500m distance was chosen since 1 km, 300m, 1-2 km are mentioned in the previous sentences. The authors have to state clearly how landscape is assumed to affect roadkill (favor large number of dears next to the road vs forces dear to cross at some specific point).
- The home range of water deer is slightly different for each study. Therefore, in this study, the home range was assumed for the water deer based on the results of the study (0.25km2) in which a tracker was attached to four water deer. The corrected part is as follows(Line 149):
“As a result of studying the behavioral rights of water deer by attaching a transmitter to four water deer, the area was 0.24km2 during the day and 0.26km2 at night [36].”
The sentence: “Similarly, the land cover near the expressway was analyzed up to an area of 500 m from the ends of the expressway to account for the water deer’s habitual range.” is also quite mysterious. What is the “habitual range”? It seems that the authors would like to make inference from the comparison of landcover along the expressway next to vs far from roadkill, but since the underlying hypothesis are not specified (e.g., no roadkill = no dear or no roadkill = no road crossing?), again it is difficult to evaluate the rational of these analyses.
- I changed "habitual range" to "home range".(Line 152)
“Based on these studies and considering the water deer’s home range, this study targeted a 500 m radius from each WVC occurrence point for analysis. Similarly, the land cover near the expressway was analyzed up to an area of 500 m from the ends of the expressway to account for the water deer’s home range.”
Equally problematic is the use of the national ecosystem survey data. We have no clue how these data are collected (opportunistic report? Systematic survey accounting for habitat dependent dear detectability?). Hence, the kind of inference that can be made for these data cannot be evaluated, as well as what the authors are expecting with the comparison between these habitat distribution data far from highway and close to highway.
- A description of the National Ecosystem Survey has been added.(Line 155)
“To analyze the land cover around the area where the water deer appeared, data from the national ecosystem survey were used. The National Ecosystem Survey, conducted by the Ministry of Environment, is the largest national natural environment survey project that targets the entire country and investigates the entire natural ecosystem, including vegetation, flora, fauna and topography.”
- The National Ecosystem Survey was used for the following study. The land cover of water deer was compared in the WVC occurrence area and the study area (Cheongju, Boeun, and Sangju). By comparing the land cover around the CSE and the area where water deer frequently appear, it was analyzed whether the area around the CSE was composed of the preferred spatial distribution of water deer. The results of this are additionally inserted in the results section.
All these spatial data are then analyzed with various methods (section 3.2 and 3.3) and I started to be highly confused. I could not follow what was the added value (specific hypothesis tested) of each of them. It seems that the authors tried any methods they knew rather than choosing the appropriate method for testing a clearly stated hypothesis. The regression analysis at the end of page 9 with a F = 239.69, R2 = 0.98, is a complete mystery (and such unusually high value are highly suspicious).
- This study was analyzed in the following order.
- WVC seasonality Modeling
Analyze the annual occurrence pattern of WVC using GAM. This analysis is conducted to analyze the relationship between the WVC occurrence pattern and the life cycle of water deer.
- Land Cover-Specific WVC Density Analysis
Using GIS, the land cover at the water deer appearance point, the land cover at the WVC, and the land cover around the CSE are analyzed.
By analyzing the land cover at the point of appearance of water deer, the land cover at the WVC point, and the land cover around the CSE, the above analysis is carried out to confirm whether the area around the CSE is the preferred area for water deer.
- WVC Hotspot Modeling
Analyze the WVC Hotspot sections of the CSE using the KDE method. And analyze the land cover of each section,
- WVC occurrence cluster analysis
By analyzing the degree of clustering in each section through the NNR method, the WVC cluster characteristics for each section are analyzed.
- We analyzed the characteristics of WVC that occurred in CSE in the above order, and analyzed the characteristics of the WVC hotspot section.
- The results of the statistics mentioned above were regression analyzed by entering land cover around CSE as the independent variable and WVC as the dependent variable. The relationship between the two resulted in a statistically significant relationship.

Reviewer 2 Report
General comments to authors:
The manuscript "A study on the characteristics of Wildlife-Vehicle Collisions in the Korean Expressways and its Hotspots" aimed (1) to predict the season when WVCs in Cheongju-Sangju Expressway (CSE) occurs most frequently, (2) to locate WVC Hotspots in CSE and to identify the landscape characteristics of Hotspot areas, and (3) to analyze WVC Hotspot and the ecological characteristics of wildlife; consequently, to establish effective WVC mitigation measures.
This research presents the results of considerable sampling effort on the part of the authors.
I believe this study is a valuable contribution to understanding the factors that influence the risk of Wildlife-Vehicle Collisions (WVCs).
Overall, the paper is well structured, and the provided information is a good resource. The article is very interesting and well written.
However, in general, some topics need improvement: (i) The introduction is partially clear. The literature review needs to help set the stage for the rest of the article, so some topics need to be explored further (see detailed comments). The study lacks clearly stated hypotheses; (ii) The analysis was thorough and well documented; however, the methods lack a better explanation (see detailed comments); (iii) Results were clearly presented; but (iv) I feel that a Discussion section with the literature can make the article better (several works were cited in the introduction and other sections and should be used to discuss the results.
Detailed comments to authors:
As the PDF doesn't have line numbers, I inserted the considerations below and in the attached PDF. Always provide a version with numbered lines, this helps the reviewer a lot.
Title:
Point 1: Remove "A study on the". Vague and imprecise term. The work itself is a study.
Point 2: I believe the title needs more focus. The title makes us think that they are wild animals in general (ie more than one group), but it is actually focused on just one species (Korean water deer; Hydropotes inermis).
Abstract:
Point 1: In what sense?
Point 2: January
Point 3: December
Introduction:
Point 1: I believe you need to better contextualize. Not only speed and traffic are important, note that factors such as traffic characteristics, land use, animal behavior and proximity of roads to the Forest can also be related to WVC.
In particular, I believe that proximity of roads to the Forest and width of roads or temporal factors greatly increase the risk of WVC.
Could you put this in a better context?
Point 2: remove the period before the citation
Point 3: The study lacks clearly stated hypotheses.
Materials and Method:
Point 1: Cheongju-Sangju (CSE)
Point 2: Would you be able to insert a photo of some stretch of road associated with the map?
Point 3: Korean water deer in mountainous areas?
Since the literature demonstrates that Korean water deer in lowland areas have different diets than those in mountainous areas, could this behavior not influence the results? And so your findings would be more restricted to Korean water deer in mountainous areas?
Perhaps this point needs to be made clear somewhere in the text.
Point 4: I believe that more information is important for purposes of similar future studies, for example:
- What was the frequency of registrations? Daily, weekly, monthly?
- Was there a standard time (in the day) for registrations? And observation/search time?
- Were the sighted animals (or carcasses?) removed from the road, not to be counted again?
I believe this information is important to facilitate replication of a given method.
Point 5: Do you mean the subspecies native to Korea (Hydropotes inermis argyropus), or the subspecies belonging to the southeast region of China (Hydropotes inermis inermis)?
Would you then be studying only the “Korean water deer”, that is, Hydropotes inermis argyropus? Or both? Make this clear.
It would also be interesting to insert an image of the species studied.
Point 6: Why? Therefore, it is necessary to describe the methods more adequately.
Point 7: Even being considered "pests" in South Korea (see Ministry of Environment 2015) due to their high density in the region and the damage that they cause to agricultural products. Why internationally is the Korean water deer considered vulnerable and is on the IUCN Red List of Threatened Species? Make that clear here.
If I'm not mistaken, it's vulnerable, right? At least in China the populations are so declining.
Hence the importance of making it clear whether the work is focusing on the two subspecies or not.
If you're only working with the Korean native subspecies, that's fine, your information is consistent with it, but if it's both, maybe not all information applies.
Point 8: Could you give a brief description for each category? Just like you did for the first one.
Point 9: Also describe in terms of percentage the intermediate sections (II-IV).
Point 10: First time quoted. Make it clear what this is about. Nearest Neighbor Method (NNM)?
Results and Discussion:
Point 1: Wildlife-Vehicle Collisions (WVCs)
Point 2: Cheongju-Sangju Expressway (CSE)
Point 3: Wildlife-Vehicle Collisions (WVCs)
Point 4: Cheongju-Sangju Expressway (CSE)
Point 5: September?
Point 6: Wildlife-Vehicle Collisions (WVC)
Point 7; Figure 4: Where are the results and discussion of this figure? Personally, I didn't see any mention of this figure in the text.
Point 8: Wildlife-Vehicle Collisions (WVCs)
Point 9: Cheongju-Sangju Expressway (CSE)
Point 10: Kernel Density Estimation (KDE).
Point 11: Also describe the nomenclature of intermediates (II-IV).
Point 12; Figure 5: Where are the results and discussion of this figure? Personally, I didn't see any mention of this figure in the text.
Point 13: Kernel Density Estimation (KDE).
Point 14: Also describe the nomenclature of intermediates (II-IV).
Point 15: You need to discuss further why the higher occurrence of WVC in Forested areas, Agricultural Lands, Grasslands and Water. It's still too vague.
Point 16: Why? There is an ecological/biological explanation. And you certainly need to make that clear. I think you need to review the subspecies you are working on (and its occurrence, mountainous) so as not to draw general conclusions about them all..
Point 17: This information seems to me contradictory.
Point 18: Maybe this should come before?
Point 19: This result is totally lacking in discussion.
Point 20: This result is totally lacking in discussion.
References:
General: Your references are not standardized correctly. Please check the "Instructions for Authors":
https://www.mdpi.com/journal/sustainability/instructions
For example, volume is in italics; journal name is abbreviated.

Author Response
First of all, thank you for taking your valuable time to review this article.
Based on the advice of the reviewers, the article was carefully revised. Thanks for the advice.
The manuscript "A study on the characteristics of Wildlife-Vehicle Collisions in the Korean Expressways and its Hotspots" aimed (1) to predict the season when WVCs in Cheongju-Sangju Expressway (CSE) occurs most frequently, (2) to locate WVC Hotspots in CSE and to identify the landscape characteristics of Hotspot areas, and (3) to analyze WVC Hotspot and the ecological characteristics of wildlife; consequently, to establish effective WVC mitigation measures.
- We have made major revisions to this paper.
- The purpose of the study has been modified as follows.
(Line # 96)
To this end, in the present study, we aim to understand the landscape characteristics of the locations frequented by water deer and the locations of WVC occurrence. Then, we identify WVC hotspots and analyze their spatial characteristics. Finally, we analyze the WVC cluster characteristics of the identified WVC hotspots. In doing so, we analyze the spatial characteristics of WVC occurrence to gather the basic data necessary for implementing WVC reduction plans on expressways.
This research presents the results of considerable sampling effort on the part of the authors. I believe this study is a valuable contribution to understanding the factors that influence the risk of Wildlife-Vehicle Collisions (WVCs).
- Thank you.
Overall, the paper is well structured, and the provided information is a good resource. The article is very interesting and well written.
- Thank you.
However, in general, some topics need improvement: (i) The introduction is partially clear. The literature review needs to help set the stage for the rest of the article, so some topics need to be explored further (see detailed comments). The study lacks clearly stated hypotheses; (ii) The analysis was thorough and well documented; however, the methods lack a better explanation (see detailed comments); (iii) Results were clearly presented; but (iv) I feel that a Discussion section with the literature can make the article better (several works were cited in the introduction and other sections and should be used to discuss the results.
- Thank you for your advice. We have made major revisions to this paper.
- Hypotheses are newly added to the introduction section
(Line # 76)
Animals are mainly found near their shelters and near locations where food is readily available. Expressways are built through wildlife habitats. Therefore, wildlife will cross any expressway built within their home range, which is the primary cause of WVCs. For this reason, it is extremely important to study the relationship between the landscape around WVC hotspots and the home ranges of wildlife.
Please see the attachment.

Reviewer 3 Report
Nice work! I recommend this manuscript for publication after consideration of the following comments.
- Please explain in the text how the results from this study can be used or generalized to the other parts of the world.
- I think that the literature review needs more work. I would recommend that you add more information related to the other similar studies to show the novelty of this work. Plus, you may need to mention why you chose "geographical information regarding animal migration based on WVC occurrences" and the advantages of this method over the others.
- In the section (2.1), add the GIS coordinates of your study area, some information that helps readers find the study site with more ease.
- I would recommend that you have "uncertainty and recommendations for future works" under results and discussion section.
Author Response
First of all, thank you for taking your valuable time to review this article.
Based on the advice of the reviewers, the article was carefully revised. Thanks for the advice.
1. Please explain in the text how the results from this study can be used or generalized to the other parts of the world.
- Thank you for your advice. We have inserted a new paragraph in the conclusion.(Line 440):
“Through this study, the occurrence of WVC is influenced by the ecological characteristics of water deer and the spatial distribution around the CSE and it was found that the WVC occurrence pattern tends to cluster spatially. Therefore, when establishing a policy to reduce WVC on a road, an effective WVC reduction plan can be established if the WVC characteristics and spatial distribution of the road are considered together.”
2. I think that the literature review needs more work. I would recommend that you add more information related to the other similar studies to show the novelty of this work. Plus, you may need to mention why you chose "geographical information regarding animal migration based on WVC occurrences" and the advantages of this method over the others.
- The following sentence has been newly inserted. (Line 116):
“The WVC data of Korea Expressway Corporation is recorded daily by employees in charge of WVC. So the data is recorded regularly. Therefore, this study used WVC data of Korea Expressway Corporation.”
3. In the section (2.1), add the GIS coordinates of your study area, some information that helps readers find the study site with more ease.
- Latitude and longitude are indicated on the map of the study area in Figure 1.
4. I would recommend that you have "uncertainty and recommendations for future works" under results and discussion section.
- We have inserted a new paragraph in the conclusion (Line 445) :
“On the other hand, this study has a limitation in that it did not consider the characteristics of various ecosystems because the study was conducted on water deer with relatively few concerns about the underreporting of WVC data and land cover was used among various spatial characteristics Therefore, if the characteristics of WVC Hotspot are analyzed in consideration of the characteristics of various ecosystems, a more reasonable WVC reduction plan will be established.”

Reviewer 4 Report
An interesting paper that adds to an understanding of how by including an analysis of topography, wildlife distribution and WVC data in roadkill mitigation measures an optimal outcome could be achieved. I believe the predictive value of your research could be in its ability to possibly predict where WVC could occur on a proposed expressway at the planning stage. Enabling mitigation measures being put in place during construction or even getting the expressway diverted would be a huge advantage in cost saving. I would suggest you implement this in your conclusion. e.g. Your final statement in the conclusion might read. Therefore, if predictable WVC hotspots are analyzed considering the characteristics of ecosystems in future expressway developments, it might be possible to put in WVC mitigation measures during the construction stage or change the direction of the proposed route.
The strength of your article is in its predictive value and the statistical analyses using several methods. A weakness is a lack of data on vehicle numbers. It is possible high WVC in Used Areas is due to more vehicles present.
There are several minor alterations which I believe would improve your paper.
Line 14. Typo. FThe.
Line 22. Used Areas. I would suggest you use the definition as stated at line 174 in the abstract, otherwise readers will not know what you are describing.
Line 31. Change 'a more reasonable' to 'an appropriate'.
Line 42. Change 'the most direct ' to 'a direct' and this statement needs a reference.
Line 58. Suggest you consider more references e.g. Road barrier effect on small birds removed by vegetated overpass in South East Queensland March 2010 Darryl Jones, Amy Blacker.
Lines 64 ---their effectiveness (AND?) establish the priority----
Lines 71/72. 'Many studies---' Statement needs a reference or two.
Lines 86/87. Rephrase. It is essential to identify spatial and temporal hotspots and analyze their effects on WVC occurrence to establish mitigation measures and thus avoid high WVC's
Lines 114. Rephrase. The CSE is located near the Songnisan National Park and so passes through abundant forests and various types of land cover.
Line 128. Typo. in in.
Line 142. Statement needs a reference.
Line 146. They have been introduced in several regions of Britain and France. This statement has no relevance to your research, suggest you remove it.
Line 289. Pearson's correlation test. I can't find this mentioned in the methods section. What are you correlating--quantify land cover? land cover type is categorical.
Line 325/326/327. You need to define time. Cases per km per day/month/year.
Line 441. Reference required for previous study.
Line 458. The results of this study----plural.
Line 464. If you do not use the suggested altered conclusion sentence then change the present one from 'a more reasonable' to 'an appropriate WVC ---'
Author Response
Please find the attached file to reply to reviewer's comments

Round 2
Reviewer 1 Report
none
Author Response
Dear Reviewer,
Thank you for your comments and we appreciate your contribution to our paper.
Best
Sangdon Lee
Reviewer 2 Report
General comments to authors:
The manuscript "A study on the characteristics of Wildlife-Vehicle Collisions in the Korean Expressways and its Hotspots" aimed (1) to predict the season when WVCs in Cheongju-Sangju Expressway (CSE) occurs most frequently, (2) to locate WVC Hotspots in CSE and to identify the landscape characteristics of Hotspot areas, and (3) to analyze WVC Hotspot and the ecological characteristics of wildlife; consequently, to establish effective WVC mitigation measures.
This research presents the results of considerable sampling effort on the part of the authors.
I believe this study is a valuable contribution to understanding the factors that influence the risk of Wildlife-Vehicle Collisions (WVCs).
Overall, the paper is well structured, and the provided information is a good resource. The article is very interesting and well written.
The authors have made efforts to amend the manuscript based upon the original comments. The changes made by the authors improve the manuscript.
Author Response
Dear Reviewer,
Thank you for your comments and we appreciate your contribution and agreement to our revised paper.
Best
Prof. Sangdon Lee
Reviewer 3 Report
I recommend this manuscript for publication.
Author Response
Dear Reviewer,
Thank you for your comments and we appreciate your contribution to our paper.
Best
Prof. Sangdon Lee
This manuscript is a resubmission of an earlier submission. The following is a list of the peer review reports and author responses from that submission.
Round 1
Reviewer 1 Report
This paper examined the wildlife-vehicle collision patterns along a section of heavily used road in Korea. I think the subject is interesting and the results have meaningful contributions for conservation. I found the paper fairly difficult to follow, however, and therefore it was difficult to evaluate and understand what was being done. The Methods, in particular, felt very incomplete. If someone were to try to replicate this study from the Methods, I don't think they could. I think more time needs to be taken so that readers can clearly understand the analytical steps that the authors took, and why.
I also found the organization of the paper somewhat confusing. The primary purpose seems to have been to locate collision Hotspots, and then to understand the landcover patterns around these. But the Methods start by talking about landcover analysis first and referring to the Hotspots as if readers will already know what this is. I think the paper should be reorganized to first introduce the concept of Hotspots, explain how these were spatially determined, and then explain the analysis done to determine patterns about them. It's also confusing how there is an analysis section for "Spatial Patterns of WVC Occurrence" and another for spatial characteristics of WVC Hotspots. How are these two things different, and why were both sets of analyses needed? Just more information and clearer explanation is needed throughout.
My specific comments are in the attached pdf.

Author Response
First of all, thank you for your advice on my article.
I briefly outlined the concept of Hotspot in the introduction (Line 71).
Meanwhile, the purpose of this study was to analyze the spatiotemporal characteristics of WVC occurrence based on WVC data, and to analyze the cluster pattern and land cover characteristics of the WVC hotspot section.
This study is:
First, we analyzed seasonal and yearly characteristics using WVCs data.
Then, the land cover at the point of occurrence of the water deer around the expressway (national ecosystem survey), and at the point of occurrence of WVC was analyzed. And the density of WVC by category of each land cover was analyzed.
And the relationship between land cover and WVC was analyzed for statistical significance through statistical analysis.
The hotspot section of CSE was derived through the KDE method and land cover for each section was analyzed. And the WVC hotspot and non-hotspot were analyzed for statistical significance.
The article has been added and revised to reflect your advice. Thank you for the advice.
=============
Please find the attached pdf including response to your comments.

Reviewer 2 Report
The authors have used wildlife-vehicle collision (WVC) data collected over a 12-year period to (1) locate hotspots along an expressway in South Korea and (2) examine relationships between hotspots and landcover near the expressway. Although these data present a wonderful opportunity for investigating potential predictors of WVCs along the expressway, this manuscript requires better organization and editing to communicate the author's work to readers. Additionally, the current organization of the manuscript seems more appropriate for a report than a peer-reviewed manuscript. For publication in a peer-reviewed journal, I would expect the described work to clearly test a hypothesis/prediction. I can imagine at least two ways in which the authors could do this with relative ease. First, given that the majority of collected data are for deer, the authors could use information about the biology/natural history of the species to predict when and where deer are most likely to cross the expressway and/or be hit by a vehicle, and then use the data to test their predictions. Second, because the dataset is so large, the authors could use a subset to examine relationships between landscape attributes and WVCs, make predictions about the likelihood of WVCs in relation to landcover, and then test the predictions with the remaining data. Either way, the manuscript would benefit from an increased focus and would present a clearer message to readers.
Specific comments are listed below:
Lines 44-46: Can the authors confirm that they have used the correct reference for 15 on line 46? I thought the statement made in lines 44-46 was interesting, but did not find evidence for it in the article referenced. Maybe I skimmed through the article too quickly?
Line 56-58: There seems to be an error in this sentence that is making it difficult to understand.
Lines 59-61: I recommend caution with the wording here. Most studies of WVC occurrence according to landscape variables establish correlations between WVC occurrence and landscape variables, but cannot establish a causal relationship.
Figure 1: Please increase the font size for the marked latitudes and longitudes. They are not currently legible. Also, "war" in the figure caption should be "was".
Lines 100-101: I think you can remove this information- it does not seem to relate to your analysis or interpretation.
Line 120: Delete "are". Also, please clarify "stay within 1 km". Do the deer stay within 1 km OF something or do they stay within a 1-km2 area?
Lines 123-124: The meaning of this sentence was unclear to me, so I looked up the reference. I think information from the reference is taken out of context and does not make sense here.
Lines 128-129: Were land cover variables averaged over years? Or were separate land cover maps used from 2008-2019 to match the years from which WVC data were collected? If land cover maps from each year were not used, can a single land cover map (or averaged land cover variables) truly represent land cover conditions throughout the entire study period? Please clarify.
Lines 132-135: Please identify your independent variable. What measure of land cover was used in the analysis?
Lines 141-143: I recommend relabeling the five WVC density categories. Referring to these categories as "sections" implies 5 distinct spatial units, which they are not.
Lines 153-154: What is meant by "regions where wildlife commonly appeared"? Please clarify.
Section 3.1: I recommend removing all information on species other than deer. While that information would be useful in a report to local agencies, it is not relevant to the focus of this study.
Lines 233-235: There seem to be some grammatical errors with this sentence(s).
Lines 243-246: Please delete the sentence on line 243-244 and simply include all results of the statistical test in parentheses following the statement on lines 245-246.
Lines 248-251: Please re-format as described above (lines 243-246).
Table 3: I recommend removing this table and instead incorporating results into parentheses within the text.
Figure 4: The caption refers to red and green colors. Were the section colors supposed to match the colors used in Figure 3? That would be much easier to interpret.
Lines 384-386: Please re-word for clarity.
Author Response
Please find the attached file including responses to your comments.

Round 2
Reviewer 1 Report
The authors reasonably addressed all concerns brought up by the reviewers.
Reviewer 2 Report
Although the authors have an interesting dataset that could be analyzed to provide useful information about the occurrences of Wildlife-Vehicle Collisions along the CS Expressway, their efforts here fall a bit short. Revisions to the manuscript have done little to demonstrate that the analytical approach employed- and interpretation of results- are appropriate. In some cases, key reviewer comments were not addressed. Other comments have been addressed with small changes in text that have done little to improve clarity.